# Immunophenotypes Based on the Tumor Immune Microenvironment Allow for Unsupervised Penile Cancer Patient Stratification

**DOI:** 10.3390/cancers12071796

**Published:** 2020-07-04

**Authors:** Chengbiao Chu, Kai Yao, Jiangli Lu, Yijun Zhang, Keming Chen, Jiabin Lu, Chris Zhiyi Zhang, Yun Cao

**Affiliations:** 1State Key Laboratory of Oncology in South China, Collaborative Innovation Center for Cancer Medicine, Sun Yat-sen University Cancer Center, Guangzhou 510060, China; chucb@sysucc.org.cn (C.C.); yaokai@sysucc.org.cn (K.Y.); lujl@sysucc.org.cn (J.L.); zhangyij@sysucc.org.cn (Y.Z.); chenkm@sysucc.org.cn (K.C.); lujb@sysucc.org.cn (J.L.); 2Department of Pathology, Sun Yat-sen University Cancer Center, Guangzhou 510060, China; 3Department of Urology, Sun Yat-sen University Cancer Center, Guangzhou 510060, China; 4Key Laboratory of Functional Protein Research of Guangdong Higher Education Institutes and MOE Key Laboratory of Tumor Molecular Biology, Institute of Life and Health Engineering, College of Life Science and Technology, Jinan University, Guangzhou 510632, China

**Keywords:** peSCC, HPV, tumor immune microenvironment, PD-L1, CTLA-4, Siglec-15, prognosis

## Abstract

The tumor immune microenvironment (TIME) plays an important role in penile squamous cell carcinoma (peSCC) pathogenesis. Here, the immunophenotype of the TIME in peSCC was determined by integrating the expression patterns of immune checkpoints (programmed cell death-1 (PD-1)/programmed cell death ligand-1 (PD-L1), cytotoxic T lymphocyte antigen 4 (CTLA-4), and Siglec-15) and the components of tumor-infiltrating lymphocytes, including CD8^+^ or Granzyme B^+^ T cells, FOXP3^+^ regulatory T cells, and CD68^+^ or CD206^+^ macrophages, in 178 patients. A high density of Granzyme B, FOXP3, CD68, CD206, PD-1, and CTLA-4 was associated with better disease-specific survival (DSS). The patients with diffuse PD-L1 tumor cell expression had worse prognoses than those with marginal or negative PD-L1 expression. Four immunophenotypes were identified by unsupervised clustering analysis, based on certain immune markers, which were associated with DSS and lymph node metastasis (LNM) in peSCC. There was no significant relationship between the immunophenotypes and high-risk human papillomavirus (hrHPV) infection. However, the hrHPV–positive peSCC exhibited a higher density of stromal Granzyme B and intratumoral PD-1 than the hrHPV–negative tumors (*p* = 0.049 and 0.002, respectively). In conclusion, the immunophenotypes of peSCC were of great value in predicting LNM and prognosis, and may provide support for clinical stratification management and immunotherapy intervention.

## 1. Introduction

Male penile squamous cell carcinoma (peSCC) is an infrequent but aggressive disease [1], and the overall incidence of peSCC is about 0.6/10^5^ in Chinese males [2]. Approximately one-third to half of all cases of peSCC are caused by persistent infection with high-risk human papillomavirus (hrHPV). hrHPV–positive (hrHPV^+^) patients tend to have better prognoses than hrHPV–negative (hrHPV^−^) peSCC patients [3]. The prognosis and survival of advanced-stage (T4 or N3) patients remains poor, due to the aggressiveness of the disease and the lack of effective systemic therapies. Immunotherapy has been demonstrated as a novel treatment modality in advanced-stage patients that have failed standard chemotherapy [4]. However, the response rate of immunotherapy is still very low, due to the scarcity of effective biomarkers linked to disease prognosis and drug response. An in-depth understanding of the tumor immune microenvironment (TIME) could better stratify cancer patients according to their prognosis, help to predict and stratify patients who will benefit from immunotherapy, and ultimately help save the lives of patients with cancer [5].

Tumor-infiltrating lymphocytes (TILs) are an important component of the TIME and are closely related to the antitumor immune response and prognosis in peSCC [6,7,8,9,10,11,12,13,14]. The increased number of FOXP3^+^ regulatory T cells (Tregs) and low inflammatory infiltrates were associated with unfavorable outcomes in peSCC [13]. There were higher numbers of CD8^+^ cytotoxic T-lymphocytes (CTLs) and FOXP3^+^ Tregs in HPV^+^ peSCC than in HPV^−^ tumors, which indicated a stronger cytotoxic immune response and immune escape in HPV^+^ peSCC [11]. This could explain, in part, why hrHPV^+^ tumors have a survival advantage. A high density of Granzyme B (GrB) was found to be associated with fewer lymph node metastases (LNM) and better survival in oral cavity carcinomas [15]. At present, however, no study has assessed the expression of GrB in peSCC.

Tumor-associated macrophages (TAMs) are a major component of the TIME and play a vital role in tumor progression. TAMs include both the M1 macrophages, which are involved in tumor suppression, and the CD206^+^ M2 macrophages, which possess tumor-promoting functions [16]. In squamous cell carcinomas (SCC), recent studies found a positive correlation between CD68^+^ macrophages (both M1 and M2 phenotypes) and tumor progression in cervical cancers and oral SCC [17,18]. High M2 TAM infiltration appeared to be positively associated with LNM rather than with a prognosis of peSCC [7]. The infiltration distribution and clinical significance of distinct macrophage subtypes in peSCC remains unclear. 

The immune checkpoints, such as programmed cell death-1 (PD-1)/programmed cell death ligand-1 (PD-L1) and cytotoxic T lymphocyte antigen 4 (CTLA-4), are known to help solid tumors evade host anti-tumor immunity. Recently, Ottenhof et al. showed that 48% of peSCC was positive for PD-L1, of which 38% had a diffuse expression associated with worse outcomes, while the prognosis of peSCC with tumor-stroma marginal PD-L1 expression was much better [14]. In head and neck squamous cell carcinoma (HNSCC), a low infiltration of PD-1 expressing lymphocytes and high CTLA-4 tumor expression was found to be significantly correlated with a poor prognosis as well as in the HPV^−^ cohort [19,20]. Lieping Chen et al. identified Siglec-15 as a novel immunosuppressive molecule [21], which may be another potential therapeutic target and prognostic indicator of peSCC. Immune checkpoint inhibitors are widely studied in cancer immunotherapy. Therefore, it is necessary to investigate the expression of PD-1/PD-L1, CTLA-4, and Siglec-15 in peSCC.

While there are many studies concerning single immune factors of the tumor environment, the complex interactions between different factors hinder the capture of comprehensive information on the TIME. Comprehensive analysis of the distribution and level of different immune markers is necessary to obtain the immunophenotype of the TIME. According to the infiltration of immune cells, the TIME was classified into three basic types: immune inflamed; immune desert; immune excluded [22]. In addition, Teng et al. proposed four different phenotypes of the TIME, based on the presence of TILs and PD-L1 expression: Type I, adaptive immune resistance (TIL^+^/PD-L1^+^); Type II, immune ignorance (TIL^−^/PD-L1^−^); Type III, intrinsic induction (TIL^−/^PD-L1^+^); Type IV, immune tolerance (TIL^+^/PD-L1^−^) [23]. Such immune phenotypes could predict the response to treatment and, hence, yield variable prognoses and outcomes.

To better recognize the immunophenotypes of the TIME in peSCC, this requires a new approach. Unsupervised clustering is a machine learning method that analyses explanatory data in an attempt to identify hidden structures in unlabeled data. This method is widely used in genomics and will give us insight into the underlying patterns of the different TIME.

Therefore, a more profound understanding of the clinical significance of the TIME requires a comprehensive analysis of multiple immune factors. Here, we investigated the expression of CD8, GrB, FOXP3, CD68, CD206, PD-L1, PD-1, CTLA-4, and Siglec-15 in 178 peSCC patients using immunohistochemistry. The patients were classified into different TIME immunophenotypes using unsupervised clustering analysis with these immune factors. We also analyzed the relationship between the TIME and hrHPV and the clinical significance of immunophenotypes.

## 2. Results

### 2.1. Clinicopathological Characteristics

Our consecutive cohort of 178 invasive peSCC patients (median age 52 years, range 24–86 years) included: 16 (9.2%) tumors with a HPV-related histological subtype; 75 (42.1%) with LNM; 20 (11.6%) with lymphovascular invasion (LVI); 27 (15.7%) with nerve invasion (NI); 20 (11.6%) with necrosis. The longest follow-up period was 209 months, and the median follow-up period was 88 months. At the end of the last follow up, 44 deaths were ascribed to peSCC. The number of patients with positive hrHPV was 58 (32.6%). HPV 16 was the most prevalent type (54/58) among the hrHPV^+^ tumors. The clinicopathologic characteristics associated with the hrHPV status are summarized in Table 1. When examining the relationships between hrHPV and other characteristics, we observed a significant difference only in death by peSCC (*p* = 0.019). The hrHPV^+^ peSCC were somewhat less differentiated compared to hrHPV^−^, although the difference was not significant (*p* = 0.081).

### 2.2. Tumor Immune Microenvironmental Characteristics Associated with hrHPV

The representative images of the expression of the immune markers in the FFPE tumor samples are shown in Appendix A. We excluded the samples that were lost during processing or did not contain any tumor epithelium from the analysis of that specific marker. As only one tissue expressed CTLA-4^+^ T cells in the tumor compartment, we noted a negligible intratumoral presence of CTLA-4^+^ T cells. Among the 178 patients with peSCC, positive PD-L1 expression on tumor cells was detected in 120 (67.4%) patients, including 80 (44.9%) that exhibited marginal expression, as shown in Figure 1a, and 40 (22.5%) that exhibited diffuse expression, as shown in Figure 1b. The PD-L1 staining in the immune cells of the stroma was positive in 94 (52.8%) patients and negative in 84 (47.2). There were 156 out of 170 cases with Siglec-15 positive expression in the stroma of the tissues, including 130 (76.5%) cases exhibiting low expression (Scores 1–3) and 26 (15.3%) exhibiting high expression (Score 4), as shown in Figure 1c,d. In the intratumoral tissues, the Siglec-15 overall positive (Score > 0) expression rate was 64.1% (109/170).

The distribution of the data between the hrHPV^+^ and hrHPV- subgroups is represented by a boxplot, as shown in Figure 2a, after the ln-transformed densities in cells/mm^2^ for CD8, GrB, FOXP3, PD-1, CTLA-4, CD68, and CD206, and by spineplot diagrams for PD-L1 and Siglec-15, as shown in Figure 2b. Compared with the hrHPV^−^ group, the hrHPV^+^ group exhibited a higher density of stroma GrB and intratumoral PD-1 (*p* = 0.049 and 0.002, respectively). The expressions of PD-L1 and Siglec-15 were not related to hrHPV infection in peSCC, as shown in Appendix A.

### 2.3. Cutoff Values for the Immune Markers Associated with Prognosis and LNM

Based on the significant differences in the immune markers between the stromal and intratumoral compartments (*p* < 0.001, paired *t*-test), we obtained cut-off values for each group. We used the “surv_cutpoint” function of the “survminer” R package to calculate the optimal cutoff value for the DSS. In the stromal region, the cutoff point was 6.28 for CD8, 5.32 for GrB, 4.94 for FOXP3, 7.25 for CD68, 4.38 for CD206, 2.68 for PD-1, and 3.06 for CTLA-4. In the intratumoral region, the cutoff point was 2.26 for CD8, 3.88 for GrB, 1.50 for FOXP3, 5.79 for CD68, 0.85 for CD206, and 0.95 for PD-1. Based on these cutoff values, we classified the patients into two groups—namely, low density vs. high density—for survival analysis. When treated as categorical variables, a high density of GrB, FOXP3, CD68, CD206, PD-1, and CTLA-4 (only except CD8) was associated with a better DSS, as shown in Appendix A.

When the densities of immune markers were analyzed as continuous variables, higher expressions of stromal FOXP3, GrB, CTLA-4, Siglec-15, and intratumoral CD206 were linked to better prognoses, as shown in Figure 3a. The patients with diffuse PD-L1 tumor cell expression had a worse prognosis than those with marginal PD-L1 expression (hazard ratio (HR) 3.055, *p* = 0.004), and those with marginal or negative PD-L1 expression (HR 2.067, *p* = 0.023), as shown in Figure 3b. However, there was no significant difference in the prognosis whether the stromal PD-L1 expression and intratumoral Siglec-15 expression were positive or not, as shown in Figure 3c,d. Patients with high Siglec-15 expression in the stroma had a better prognosis, as shown in Figure 3e. In the Cox multivariate regression model using the densities of immunomarkers as categorical variables, diffuse PD-L1 expression (HR 2.382, *p* = 0.025), high stromal FOXP3 (HR 0.366, *p* = 0.006), and high intratumoral PD-1 (HR 0.452, *p* = 0.016) were independent prognostic factors for survival after adjusting the clinical parameters, as shown in Appendix A. When treated as continuous variables, the diffuse PD-L1 expression (HR 1.864, *p* = 0.092), high stromal FOXP3 (HR 0.737, *p* = 0.013), and high intratumoral CD206 (HR 0.806, *p* = 0.062) were independent prognostic factors, as shown in Appendix A. Overall, the densities of certain immune markers were significantly associated with prognosis in peSCC, whether treated as categorical variables or as continuous variables.

We also identified that intratumoral GrB^+^ lymphocytes, stromal CD68^+^ macrophages and CTLA-4^+^ lymphocytes were inversely associated with LNM. In the multivariable model, low intratumoral GrB^+^ (odds ratio (OR) 0.577, *p* = 0.012) and stromal CTLA-4^+^ lymphocytes (OR 0.659, *p* = 0.016) remained independent predictors for LNM, as shown in Appendix A.

### 2.4. Immunophenotypes in peSCC

The relationship between the different immune markers was explored by Spearman correlation coefficient tests, as shown in Figure 4a. The levels of each immune marker expressed in the stroma and tumors were positively correlated (*p* < 0.001). FOXP3+ Tregs were significantly correlated with CD8^+^/GrB^+^ CTLs, PD-1^+^, and CTLA-4^+^ T cells. PD-L1 tumor cell expression was positively correlated with CD8^+^/GrB^+^ CTLs, FOXP3^+^ Tregs, PD-1^+^ T cells, and CD68^+^ TAMs. We observed that PD-L1 was co-expressed on CD8^+^, CTLs, and CD68^+^ macrophages in dual immunohistochemistry (IHC) staining, as shown in Appendix A. Siglec-15 expression was significantly increased in PD-L1^+^ tumor cells and CD206^+^ TAMs.

The immune spectrum generated from the expression of multiple immune markers provided a more complete picture of a patient’s immune status. To group patients based on the similarities of immunophenotypes, we performed unsupervised hierarchical clustering on heatmaps, as shown in Figure 4b. An elbow plot analysis from one to ten clusters indicated that four clusters captured an appropriate number of the segmentation of patients with peSCC, as shown in Appendix A. The clustering analysis included 163 patients who were divided into four groups. Clusters A, B, C, and D included 30 (18.4%), 24 (14.7%), 27 (16.6%), and 82 (50.3%) patients, respectively. Fifteen patients failed to be in any cluster due to missing values. Clusters A, B, C, and D were termed immunophenotypes or immune clusters because they were stratified according to the immune parameters.

We then evaluated the expression of major immunohistochemical markers across each hierarchical cluster, as shown in Figure 4c. In cluster A, the expression of most markers (CD8, GrB, FOXP3, CD68, CD206, and PD-1) decreased, though the density of CTLA-4 was not different when compared with other subtypes. In cluster B, however, the opposite was found; penile tumors had massive lymphocyte and macrophage infiltration. In cluster C, there was rare CD206^+^ TAMs intratumoral infiltration despite a large amount of lymphocyte and CD68^+^ macrophage infiltration. In cluster D, the tumors exhibited medium levels of lymphocyte and macrophages infiltration.

For PD-L1, the proportions of diffusely PD-L1 positive tumors were 6.7% (2/30), 54.2% (13/24), 29.6% (8/27), and 18.3% (15/82) from clusters A to D, respectively, as shown in Appendix A. Cluster B had extremely high tumor cell PD-L1 diffuse expression among the four clusters (*p* < 0.001). In the stromal compartment, 0% (0/30), 91.7% (22/24), 63.0% (17/27), and 58.5% (48/82) of patients in clusters A to D demonstrated PD-L1 expression, respectively (*p* < 0.001). The frequency of stromal PD-L1 expression was the highest for cluster B and the lowest for cluster A. For Siglec-15, the proportions of intratumoral positivity were 33.3% (10/30), 83.3% (20/24), 59.3% (16/27), and 73.2% (60/82) from clusters A to D, respectively. Cluster D expressed higher levels of stromal Siglec-15 positive cells (98.8%, 81/82) than cluster A (*p* < 0.05).

In short, immune cluster A was characterized by a deficiency of TILs and TAMs in either the tumors or the stroma with no PD-L1 expression, as shown in Appendix A, and was called the “immune ignorance” or “cold tumors” immunophenotype. Immune cluster B was characterized by high levels of intratumoral CD8^+^ CTLs infiltration with high PD-L1 expression, as shown in Appendix A; thus, this was named the “immune inflammatory” immunophenotype. Immune cluster C, which was broadly populated with immune cells but relatively void of CD8^+^ CTLs and CD206^+^ macrophages, as shown in Appendix A, in the tumor core, was termed the “immune exclusion” immunophenotype. Immune cluster D had moderate levels of lymphocytes and macrophage infiltration with high stroma Siglec-15 expression, as shown in Appendix A, and was named the “immune tolerance” immunophenotype. The role of immune markers in tumor progression and the expression levels of immune markers in each immune cluster are summarized in Appendix A, respectively.

### 2.5. Immunophenotypes Associated with Prognosis

We examined the prognostic impact of the immunophenotypes using Kaplan–Meier analysis and log-rank tests. Clusters A and C were associated with worse prognoses, as shown in Figure 4d, and a similar association was also observed when stratifying for hrHPV^−^, as shown in Figure 4e, and hrHPV^+^, as shown in Figure 4f, cases separately. To refine our observation, we plotted patient survival according to clusters A and C vs. clusters B and D, and confirmed a clear and significantly worse prognoses for patients in clusters A and C, as shown in Figure 4g, who were thereby defined as high-risk clusters, while clusters B and D were defined as low-risk clusters.

We also analyzed the prognostic impact of the immunophenotypes while considering other prognostic factors. Univariate factors associated with the DSS were identified by a multivariate Cox proportional hazard analysis, as shown in Table 2. The T stage (*p* = 0.004, HR = 1.709 (1.181–2.472)), N stage (*p* < 0.001, HR = 30.403 (7.212–128.174)), hrHPV (*p* = 0.012, HR = 0.359 (0.162–0.797)) and immunophenotypes (*p* = 0.014, HR = 2.349 (1.191–4.633)] were identified as independent prognostic factors for the DSS. Indeed, the c-statistic was 0.861 for the model with T stage, N stage, and hrHPV, and was 0.875 after incorporating the immunophenotypes. Thus, the identification of the immunophenotypes is of great significance for explaining peSCC survival.

### 2.6. Association of Immunophenotypes with LNM

We further investigated the association of the immunophenotypes with well-known clinicopathological features (age, grade, histological subtypes, T stage, LNM, and hrHPV status) in peSCC, as shown in Table 3. 

No statistically significant difference among the four groups for hrHPV infection was found. Cluster A had a greater proportion of advanced T stages (T3 or T4) than the other three clusters (33.3% vs. 9.3%, *p* = 0.001), as shown in Figure 5a. The incidence of LNM was significantly higher in clusters A and C than in clusters B and D (60 and 59.3 vs. 45.8 and 32.9%, *p* = 0.021), as shown in Figure 5b. 

In multivariable analysis, as shown in Table 4, the immunophenotypes remained significantly related to LNM (*p* = 0.037, OR = 2.482 (1.057–5.833)). The results indicate that patients in cluster A were frequently accompanied with advanced T and N stages.

## 3. Discussion

The tumor immune microenvironment plays an important role in the pathogenesis of peSCC. This is the first study, to our knowledge, to report the expression of GrB, PD-1, CTLA-4, and Siglec-15 in peSCC and to propose four immunophenotypes with relevance to the prognosis and LNM, based on multiple immune parameters.

CD8^+^ CTLs are primarily responsible for killing tumor cells, while the increase in Tregs is thought to inhibit host immunity and promote tumor progression. In addition, GrB is a cytotoxic molecule secreted from activated CTLs and natural killer cells that induce neoplastic cell apoptosis [24]. As expected, the high expression of GrB was related to better outcomes in peSCC. The CTLA-4 and PD-1 immune checkpoint pathways downregulated T cell activation, thereby promoting immune tolerance and inhibiting anti-tumor immunity [25]. Despite the definitive immunosuppressive roles of CTLA-4 and PD-1, their prognostic impacts have been controversial to date. Indeed, the observation of better prognoses in patients with high densities of PD-1^+^, CTLA-4^+^ T cells, and FOXP3^+^ Tregs was counter-intuitive. The results contrasted with previous studies in peSCC and other tumor types [13].

There may be several reasons for this discrepancy. The expression of CTLA-4 and PD-1 reflects the activation of T cells, but only indicates that T cells have received an immunosuppressive signal, not that the T cells must be depleted and dysfunctional. Our results indicate that, despite the varying functional status, grouping CTLA-4^+^ or PD-1^+^ cells is limited. CTLA-4 exhibited a linear positive correlation with CD8^+^ CTLs and FOXP3^+^ Tregs, due to CTLA-4 being predominantly expressed in FOXP3^+^ Tregs and newly activated T cells [26]. FOXP3^+^ Tregs were observed to be positively related to CD8^+^/GrB^+^ CTLs, and PD-1^+^ and CTLA4^+^ T cells, suggesting that the upregulation of FOXP3, PD-1, and the CTLA-4 expression of T cells could be the result of increased immune infiltration.

We found that patients with diffuse PD-L1 tumor expression had significantly poorer prognoses than those with marginal expression. The expression of marginal PD-L1 may be induced by extrinsic factors, such as interferon gamma (IFNγ), tumor necrosis factor α (TNFα), and interleukin-1β (IL-1β), produced locally by activated T lymphocytes in the stroma [9,27]. In contrast, diffuse expression may be caused by genetic abnormalities, such as phosphatase and tensin homolog (PTEN) loss and aberrant Janus kinase-signal transducer and activator of transcription (JAK/STAT) signaling [28]. This indicates that the marginal positive reaction reflects an active immune response and has a correspondingly good prognosis, while the diffuse positivity represents patients with a high level of genetic mutations [14,28]. We also noted that not all patients with diffuse PD-L1 tumor expression had poor survival, such as peSCC with high intratumoral CD8^+^ CTLs infiltration. This indicated that the progression and prognosis of peSCC may result from a combination of immune factors.

Hence, to provide a more comprehensive analysis of the TIME in peSCC, immunophenotypes based on multiple immune factors become necessary. Through unsupervised clustering, we identified four clusters of patients with peSCC. These immunophenotypes: (i) were related to varying immune factor infiltrations and specific tumor immune microenvironments; (ii) provided an independent prognostic indicator; (iii) provided a significant predictor for LNM.

Immune cluster A was characterized by a deficiency of TILs and TAMs in either the tumor or the stroma with no PD-L1 expression. Therefore, this group was called “immune ignorance” or “cold tumors” immunophenotype. Immune ignorance was associated with an advanced T stage, LNM, and had the poorest disease prognosis among the TIME subgroups. This may be attributed to the lack of the expression of activation markers and rejection by tumors, where adaptive immunity cannot recognize or respond to pathogens or malignant tumors [29]. Studies have shown that a single-agent checkpoint blockade (anti-PD-L1/PD-1) is rarely effective in this subgroup [22]. However, the combination of anti-PD-1 and anti-CTLA-4 might improve the therapeutic effects [23].

The most representative pathologic feature of cluster B was the high levels of intratumoral CD8^+^ cytotoxic T-cell infiltration with high PD-L1 expression; thus, this was named the “immune inflammatory” immunophenotype. CD8^+^, TILs, and PD-L1 often showed a positive correlation, indicating that an adaptive immune resistance mechanism may be occurring. Deng et al. found that CD8^+^ TILs could induce PD-L1 expression in tumors through the production of IFNγ [9]. This suggested the presence of preexisting intratumoral T cells that were likely inhibited by PD-L1. The research showed that the clinical response to anti-PD-L1/PD-1 therapy was most common in patients with immune resistant tumors [23]. Therefore, being able to accurately identify this subgroup may also be beneficial for anti-PD-L1/PD-1 therapy, and for avoiding the additional potential toxicity and cost of using combined immunotherapy in peSCC.

Immune cluster C, which was broadly populated with immune cells but relatively void of CD206^+^ macrophages in the tumor core, was termed as the “immune exclusion” immunophenotype. This group was found to be related to LNM and to a poor clinical prognosis in peSCC. Interestingly, the high numbers of CD68^+^ and CD206^+^ TAMs were associated with improved prognoses in our cohort. In contrast, several other studies found that a high level of CD206^+^ TAMs was positively correlated with shorter survival among HNSCCs and hepatic carcinoma [30,31]. Several hypotheses could explain this paradoxical favorable prognosis of CD206 expression on TAMs. While CD206 is a recognized M2 macrophage marker, low CD206 expression can be found on dendritic cells and specific lymphatic or endothelial cells [30]. CD206-expressing TAMs might include tumor infiltration immune cells that exert anti-tumor effects in the tumor microenvironment. In fact, CD206 participates in antigen presentation, endocytosis and phagocytosis, signal transduction, innate host defense, and adaptive immune response, which indicates that CD206 could play both pro-tumor and anti-tumor roles [31]. In addition, the conflicting prognosis of high CD206 expression could be the result of increased immune infiltration in the TIME. A preclinical study demonstrated that CD206^+^ macrophages were a potential anti-tumor therapeutic target [32].

Immune cluster D had moderate levels of lymphocyte and macrophage infiltration with high stroma Siglec-15 expression and was named the “immune tolerance” immunophenotype. This group comprised relatively early tumor stages and was associated with good prognoses. Siglec-15 is widely upregulated on various tumor cells and tumor-infiltrating myeloid cells, as it is normally only expressed on certain myeloid cells and its expression is mutually exclusive to PD-L1 in non-small-cell lung carcinomas (NSCLCs). Siglec-15 was negatively regulated by macrophage colony-stimulating factor (CSF) and IFNγ, thus inhibiting antigen-specific T cell responses and promote tumor growth [21]. It is curious that the high expression of Siglec-15 correlated with good prognosis. In this study, Siglec-15 expression was significantly related to the CTL granules, GrB. This might explain why the high Siglec-15 expression in peSCC tended to be associated with better prognoses. However, the expression and prognostic significance of Siglec-15 in human tumors clearly warrants further investigation. A current ongoing clinical trial by Chen et al. is evaluating the effect of an anti-human Siglec-15 mAb (NC318) on solid tumors [33], which may bring new hope for patients.

Although we found no difference in the hrHPV infection rates among the four immunophenotypes, hrHPV^+^ peSCC had a higher density of stroma GrB and intratumoral PD-1 positive T lymphocytes. The tumor microenvironment of hrHPV^+^ peSCC was somehow different from that of hrHPV^−^ cancer. HPV–positive HNSCCs were much more strongly immune-infiltrated compared with HPV–negative tumors, with higher levels of T cell infiltration and overall immune cell infiltration [34]. Consistent with higher infiltration by CD8^+^ CTLs, HPV^+^ tumors harbored higher levels of immune activation—specifically perforin, Granzyme A, and Granzyme B expression. Interestingly, the better outcomes seen in HPV^+^ tumors may be partially related to their higher immune infiltration, as we observed. This may be explained by the highly immunogenic property of hrHPV, which could recruit immune effector cells, and upregulate the PD-1 and CTLA-4 immunosuppressive pathways [35]. Our findings also implicated mechanisms of T-cell escape in hrHPV^+^ peSCC, wherein a high tumoral HPV-antigen load resulted in the high expression of PD-1 on T-cells. However, the mechanisms by which HPV infection affects the immune checkpoint pathways are yet to be fully defined. 

There were some limitations worth addressing in the present study. First, it was a single-center retrospective study. In some cases, several immunohistochemical tests were not available due to insufficient tissue materials. Second, we chose CD68 and CD206 to label macrophages and M2 macrophages, respectively. There are also alternative immune molecular markers, such as CD163, CD11, etc., for labeling M2 macrophages, which may lead to some differences in the results. Third, we used digital high-resolution images and professional computer software to evaluate immune markers objectively and quantitatively, thus limiting the subjectivity of the observer. However, there was no uniform standard for the cutoff value of immune markers, such as the mean or median, and we used the cutoff value for the DSS as the cutoff value. Fourth, we did not perform multiple immunofluorescence tests on all the samples to label and locate specific immune cells in the TIME. External validation is required in other populations to confirm these study findings.

## 4. Materials and Methods

### 4.1. Patient Selection and Clinical Data Collection

The study included all men with peSCC admitted to the Sun Yat-sen University Cancer Center (SYSUCC) (Guangzhou, China) between January 1999 and December 2013. One hundred and seventy-eight consecutive patients diagnosed with peSCC were eligible with the following criteria: (1) histologically confirmed diagnosis of invasive peSCC; (2) sufficient formalin-fixed, paraffin-embedded (FFPE) block with available primary tumor tissue for this study; (3) no other tumor occurrences; (4) availability of the patient’s complete clinical history and pathological diagnostic data.

For each patient, their clinicopathological information (their age, histological subtype, grade, staging (the eighth edition of the American Joint Committee on Cancer tumor-node-metastasis (TNM) system), presence of lymphovascular and nerve invasion, and necrosis ratio) and follow-up data were retrospectively reviewed. The tumor histology subtypes were classified as HPV-related or non-HPV-related carcinomas in whole tissue sections, according to the World Health Organization (WHO) 2016 classification of peSCC [36]. The study was approved by the Ethics Committee of Sun Yat-sen University Cancer Center.

### 4.2. hrHPV Detection and Typing

The hrHPV status was determined on FFPE tissue samples using real-time polymerase chain reaction (PCR) with a Hybribio Assay (HybriMax, Chaozhou Hybribio Limited Corp., Chaozhou, China) that detected the 23 HPV types [37]. The kit could identify 13 high-risk HPVs (HR-HPVs) (subtypes 16, 18, 31, 33, 35, 39, 45, 51, 52, 56, 58, 59, and 68), five low-risk HPVs (LR-HPVs) (subtypes 6, 11, 42, 43, and 44), and five other HPV types (subtypes 53, 66, 73, 82 and 81/CP8304).

### 4.3. Immunohistochemistry and Scoring

The representative FFPE tissue specimens were sectioned continuously at four micrometers thick and stained for immunohistochemistry. Staining was performed following the manufacturer protocols. The details of the different IHC staining processes are summarized in Table 5.

For CD8, GrB, FoxP3, CD68, CD206, PD-1, and CTLA-4 analysis, we used the Aperio ImageScope (Leica Biosystems, Solms, Germany) to randomly select three peripheral and three central tumor focus fields magnified by 200× in each sample. Necrotic areas representing nonspecific inflammatory infiltration were excluded. The inForm 2.1 Image Analysis Software (Mantra Software/PerkinElmer) was used to analyze the captured images, which included the intratumoral and peritumoral stromal regions, as shown in Figure 6a,e. Briefly, tissue segmentation of the images was performed, where the tumor (in red) and stroma (in green) were distinguished and labeled, as shown in Figure 6b,f. Once each region of the tissue was defined, we determined the total number of positive immune cells within the tumor and stromal areas, as shown in Figure 6c,g. Finally, the total cell counts for each marker were divided by the total areas of each region in order to determine the final densities for each immune marker in both the intratumoral and stromal regions, as shown in Figure 6d,h.

For PD-L1 staining, >1% tumor cell membranous staining was considered as positive, and the PD-L1 expression of immune cells in the stroma was binarily scored (negative or positive) [14]. We evaluated the tumor cell expression patterns of PD-L1, including the diffuse (throughout the whole tumor fields) or marginal (the peripheral staining at tumor–stroma margin) expression [14,28]. We also selected five PD-L1^+^ tissues for CD8/PD-L1 and CD68/PD-L1 dual staining. PD-L1 staining was first performed using the avidin–biotin horseradish peroxidase (HRP) technique. After color development, the 3,3′Diaminobenzidine tetrahydrochloride substrate (DAB) was removed, and the sections were washed with water and phosphate-buffered saline (PBS). Then, the CTLs marker, CD8, or TAMs marker, CD68, was labelled using streptavidin–biotin–alkaline phosphatase complex (Dako, Glostrup, Denmark) system with vector red as a chromogen.

For estimating the Siglec-15 expression, three high fields (magnification, ×400) containing the most stained Siglec-15 (‘hotspots’) were assessed using an Olympus BX53 microscope (Olympus Corporation, Tokyo, Japan). Then, the mean numbers of intratumoral and stromal positive cells were counted in the hotspots. The tumor and stroma were separately scored as follows: 0—<1 cells/high magnification field; 1—1 to 5 cells/high magnification field; 2—5 to 20 cells/high magnification field; 3—20 to 50 cells/high magnification field; 4—>50 cells/high magnification field. 

All the results were evaluated and checked by two pathologists (Chu CB and Lu JL) who were blinded to the patients’ clinical data.

### 4.4. Statistical Analysis

Statistical analyses were performed using SPSS (version 24, IBM, New York, NY, USA) and R software (version 3.3.0, Vienna, Austria). The figures were created using R software. The normality and log-normality of the distributions of continuous variables were examined using the Shapiro–Wilk test. The immune cell counts with a lognormal distribution were transformed using a natural logarithm (ln). The optimal cutoff points of the expressions of CD8, GrB, FOXP3, CD68, CD163, PD-1, and CTLA-4 were determined by the “surv_cutpoint” function of the “survminer” R package. Accordingly, the patients were divided into a low- or high-expression group, with the optimal cutoff point for each immune marker serving as the cut-off value. Comparisons between groups were performed using Fisher’s exact test or the Chi-square test for the categorical variables and Student’s *t*-test or the Kruskal–Wallis test for continuous variables.

The association between different variables was performed with the Spearman rank correlation analysis. The package ggcorrplot was used for visualizations. To consider the complexity of the TIME of a large number of immune markers, we used a statistical method of unsupervised clustering, as previously used in genomic research. The clusters of the immunophenotypes were performed with the help of the R package pheatmap v1.0.12, using correlation as the clustering distance and ward.D as the linkage. Clusters were identified using the cutree function. To determine the optimal number of clusters, we performed elbow analysis of the KMeans using the “factoextra” R package.

The disease-specific survival (DSS) was calculated from the date of diagnosis until disease-caused death or the end of the follow-up. The Kaplan–Meier method and the log-rank test were used in the univariate analysis. Variables with *p* < 0.10 in the univariate analysis were selected for the multivariable backward stepwise Cox regression to build prognostic models. The c-statistic was used to assess the models’ goodness of fit. The predictive value of LNM was investigated using a logistic regression analysis. The sensitivity and specificity of the multivariate logistic regression models were measured using the area under the receiver operating characteristic (ROC) curve (AUC). The results were considered significant for *p*-values < 0.05.

## 5. Conclusions

In conclusion, peSCC grows in a very complex microenvironment, which can affect tumor progression and prognosis. We thereby proposed four immune-related clusters of the TIME in peSCC, which can improve prognosis stratification and provide new insights for the development of therapeutic treatments in the future.

## Figures and Tables

**Figure 1 cancers-12-01796-f001:**
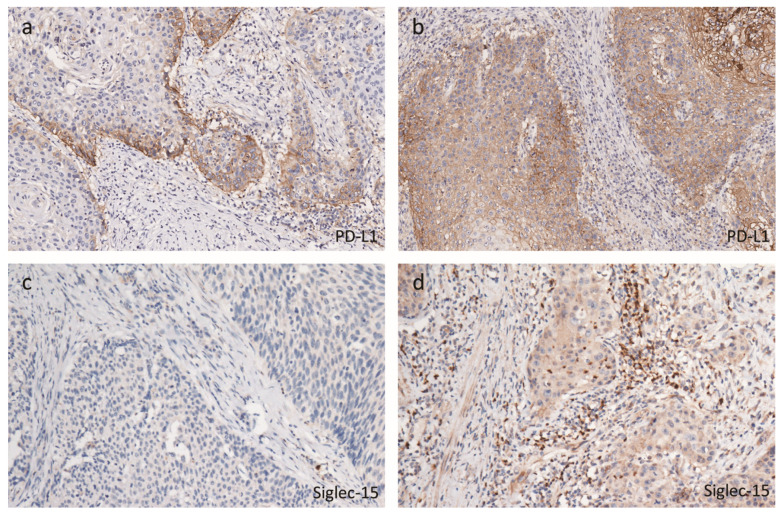
The expression of PD-L1 and Siglec-15 in peSCC tissues by immunohistochemistry. Representative immunohistochemistry (IHC) images show marginal (**a**) and diffuse (**b**) expression of PD-L1 expression (**b**). The low (**c**) and high (**d**) expressions of Siglec-15 in intratumoral or stromal tumor-infiltrating myeloid cells were also presented. Magnification: 200×.

**Figure 2 cancers-12-01796-f002:**
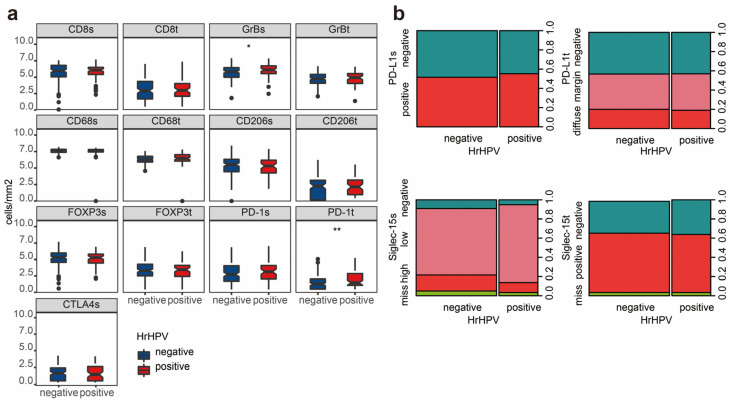
The expression of immune markers in hrHPV^−^ and hrHPV^+^ samples. (**a**) The box plots indicate the transformed densities of stromal CD8 (CD8s), intratumoral CD8 (CD8t), GrBt, GrBs,CD68t, CD68s, CD206t, CD206s, FOXP3t, FOXP3s, PD-1t, PD-1s and CTLA-4s. Compared with the hrHPV^−^ patients, the hrHPV^+^ patients expressed a higher density of stromal GrB and intratumoral PD-1. * *p* < 0.05, ** *p* < 0.01, independent t-test. (**b**) Spineplot diagrams show the expression patterns of PD-L1 and Siglec-15 in the intratumoral and stromal regions. There was no difference between the expression of PD-L1 and Siglec-15 between hrHPV^−^ and hrHPV^+^ tumors.

**Figure 3 cancers-12-01796-f003:**
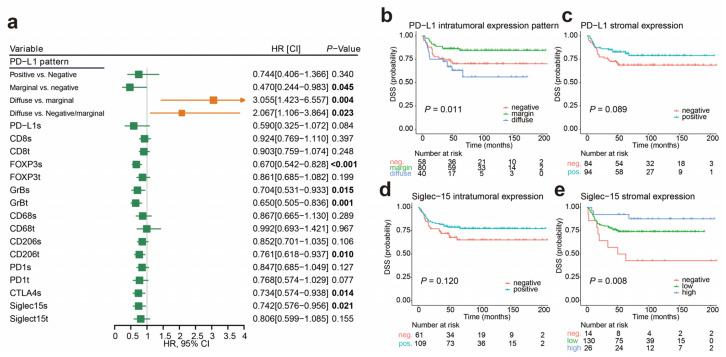
The correlation of patient prognosis and expression of immune markers in peSCC. (**a**) The densities of immune markers were treated as continuous variables. Forest plot shows the hazard ratios (HR) for disease-specific survival (DSS). Values shown in bold are statistically significant. (**b**–**e**) Kaplan–Meier survival analyses (log-rank tests) were conducted, according to PD-L1 expression in tumor (**b**), PD-L1 expression in stroma (**c**), Siglec-15 expression in tumor (**d**), and Siglec-15 expression in stroma (**e**).

**Figure 4 cancers-12-01796-f004:**
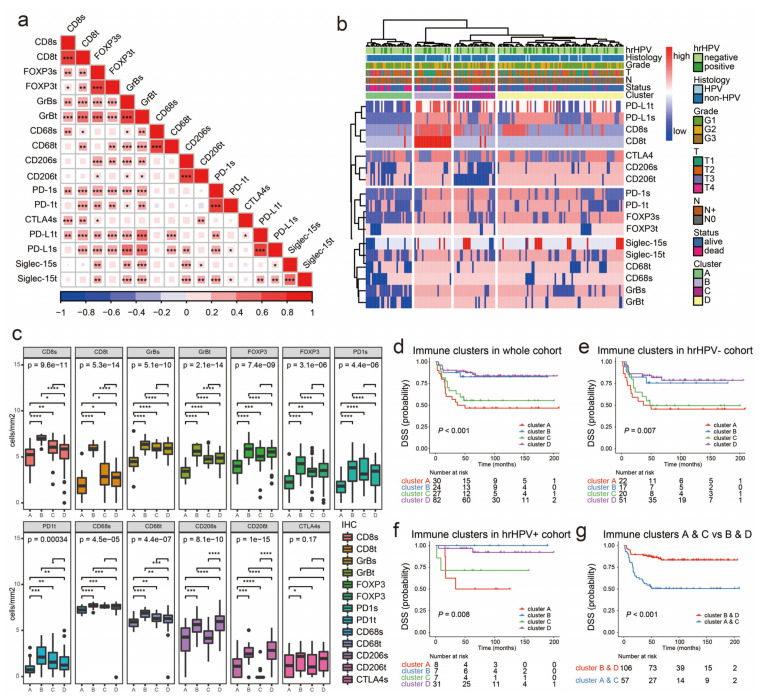
The immunophenotypes of the tumor immune microenvironment (TIME) in peSCC. (**a**) Correlation matrix plots by Spearman correlation analyses show pairwise positively stronger correlation of the expression of tested markers. * *p* < 0.05, ** *p* < 0.01, *** *p* < 0.001. (**b**) Cluster analyses using correlation distance and ward.D were conducted based on the expression of immune markers. The heatmap shows the high expression (red) or low expression (blue) of each immune marker. Annotations of the samples on the top of the heatmap indicate histopathological features, including hrHPV status, histological subtypes, grade, T stage, N stage, survival status, and the four clusters identified by the cutree method. (**c**) Box plots show the expression of immune markers, according to identified four clusters. The transformed densities of CD8, GrB, FOXP3, PD-1, CD68, CD206, and CTLA-4 in the intratumoral and stromal regions were indicated by box plots. * *p* < 0.05, ** *p* < 0.01, *** *p* < 0.001, **** *p* < 0.0001, Kruskal-Wallis test. (**d–f**) Kaplan–Meier survival curves for disease-specific survival (DSS) show patients in clusters A and C had better prognosis than clusters B and D in all cases (d), hrHPV^−^ cases (e) and hrHPV^+^ cases(f). (**g**) Patients in high-risk group (clusters A and C) were likely to experience tumor progression, compared with those in low-risk group (clusters B and D).

**Figure 5 cancers-12-01796-f005:**
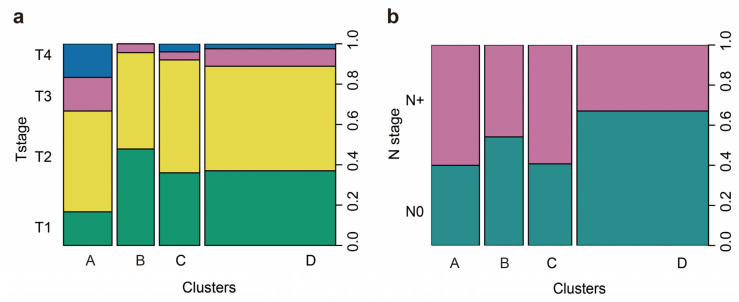
The association of immunophenotypes and disease stages of patients with peSCC. Spineplot diagrams show the percentages of patients at different T (**a**) and N (**b**) stages. Results indicate that patients in cluster A were frequently accompanied with advanced T and N stages.

**Figure 6 cancers-12-01796-f006:**
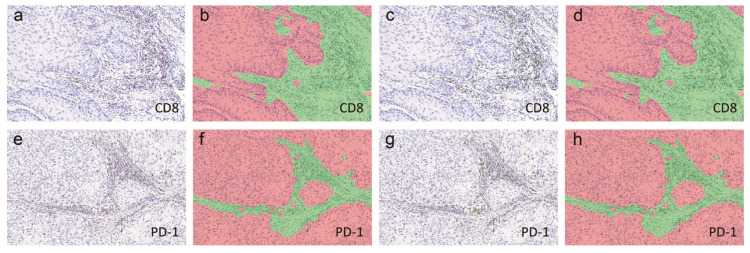
Immunohistochemical quantitative analyses of CD8 (**a**–**d**) and PD-1 (**e**–**h**) expression in intratumoral and stromal regions. Analyses were performed using inForm 2.1 Image Analysis Software (Mantra Software/PerkinElmer). Images of IHC staining, as well as the simulative tissue segmentation (tumor: red, stroma: green) (**b**,**f**), positive cell segmentation (**c**,**g**) and merged images (**d**,**h**), were shown. Magnification: 200×.

**Table 1 cancers-12-01796-t001:** Clinicopathological characteristics associated with hrHPV status.

Variable	Total, *n* = 178 (%)	HrHPV^−^, *n* = 120 (%)	HrHPV^+^, *n* = 58 (%)	*p*
**Age**				
Median (range)	52.0 (24.0–86.0)	52.5 (26.0–79.0)	51.5 (24.0–86.0)	0.871
**Grade of differentiation**				0.081
G1	106 (61.3)	77 (67.0)	29 (50.0)	
G2	53 (30.6)	31 (27.0)	22 (37.9)	
G3	14 (8.1)	7 (6.1)	7 (12.1)	
**T stage**				0.255
T1	66 (37.9)	46 (39.3)	20 (35.1)	
T2	85 (48.9)	56 (47.9))	29 (50.9)	
T3	14 (8.0)	7 (6.0)	7 (12.3)	
T4	9 (5.2)	8 (6.8)	1 (1.8)	
**N stage**				0.856
N0	103 (57.9)	70 (58.3)	33 (56.9)	
N+	75 (42.1)	50 (41.7)	25 (43.1)	
**Histological subtype**				0.143
HPV-related	16 (9.2)	8 (7.0)	8 (13.8)	
Non HPV-related	157 (90.8)	107 (93.0)	50 (86.2)	
**M stage**				0.817
M0	171 (96.1)	115 (95.8)	56 (96.6)	
M1	7 (3.9)	5 (4.2)	2 (3.4)	
**LVI**				0.256
No	152 (88.4)	103 (90.4)	49 (84.5)	
Yes	20 (11.6)	11 (9.6)	9 (15.5)	
**NI**				0.351
No	145 (84.3)	94 (82.5)	51 (87.9)	
Yes	27 (15.7)	20 (17.5)	7 (12.1)	
**Necrosis**				0.527
No	152 (88.4)	102 (89.5)	50 (86.2)	
Yes	20 (11.6)	12 (10.5)	8 (13.8)	
**Death by penile cancer**				**0.019**
No	134 (75.3)	84 (70.0)	50 (86.2)	
Yes	44 (24,7)	36 (30.0)	8 (13.8)	

G1: Well-differentiated, G2: moderately-differentiated, G3: Poorly-differentiated, LVI: lymphovascular invasion, NI: nerve invasion; Significant values (*p* < 0.05) are indicated as bold.

**Table 2 cancers-12-01796-t002:** Multivariate Cox regression analysis of immune clusters associated with DSS.

Variable	Contrast	Multivariable Analysis	
		HR [95% CI]	*p*
T stage	Per T stage	1.709 [1.181–2.472]	0.004
N stage	N+ vs N0	30.403 [7.212–128.174]	<0.001
hrHPV status	Positive vs. Negative	0.359 [0.162-0.797]	0.024
Immune clusters	A and C vs. B and D	2.349 [1.191-4.633]	0.014

HR: hazard ratio.

**Table 3 cancers-12-01796-t003:** Clinicopathological characteristics associated with immune clusters.

Variable	Total, *n* (%)	Cluster A, *n* (%)	Cluster B, *n* (%)	Cluster C, *n* (%)	Cluster D, *n* (%)	*p*
**Age**	51.9 ± 13.61	48.37 ± 14.77	51.13 ± 13.52	53.93 ± 16.31	52.76 ± 12.16	0.391
**Grade of differentiation**						0.066
G1	95 (60.1)	19 (65.5)	12 (52.2)	12 (46.2)	52 (65.0)	
G2	50 (31.6)	8 (27.6)	7 (30.4)	9 (34.6)	26 (32.5)	
G3	13 (8.2)	2 (6.9)	4 (17.4)	5 (19.2)	2 (2.5)	
**T stage**						**0.013**
T1 and T2	137 (86.2)	20 ^a^ (66.7)	22 ^a,b^ (95.7)	23 ^a,b^ (92.0)	72 ^b^ (88.9)	
T3 and T4	22 (13.8)	10 ^a^ (33.3)	1 ^a,b^ (4.3)	2 ^a,b^ (8.1)	9 ^b^ (11.1)	
**N stage**						**0.020**
N0	91 (55.8)	12 ^a^ (40.0)	13 ^b^ (54.2)	11 ^a^ (40.7)	55 ^b^ (67.1)	
N+	72 (44.2)	18 ^a^ (60.0)	11 ^b^ (45.8)	16 ^a^ (59.3)	27 ^b^ (32.9)	
**Histological subtype**						0.421
HPV-related	16 (10.1)	1 (3.4)	4 (17.4)	3 (11.5)	8 (10.0)	
Non HPV-related	142 (89.9)	28 (96.6)	19 (82.6)	23 (88.5)	72 (90.0)	
**M stage**						0.557
M0	156 (95.7)	28 (93.3)	22 (91.7)	26 (96.3)	80 (97.6)	
M1	7 (4.3)	2 (6.7)	2 (8.3)	1 (3.7)	2 (2.4)	
**LVI**						0.282
No	138 (87.9)	24 (82.8)	22 (95.7)	20 (80.0)	72 (90.0	
Yes	19 (11.6)	5 (17.2)	1 (4.3)	5 (20.0)	8 (10.0)	
**NI**						0.563
No	131 (83.4)	24 (82.8)	21 (91.3)	19 (76.0)	67 (83.8)	
Yes	26 (16.6)	5 (17.2)	2 (8.7)	6 (24.0)	13 (16.2)	
**Necrosis**						0.597
No	138 (87.9)	25 (86.2)	19 (82.6)	21 (84.0)	73 (91.2)	
Yes	19 (12.1)	4 (13.8)	4 (17.4)	4 (16.0)	7 (8.8)	
**Death by penile cancer**						**<0.001**
No	114 (73.0)	14 ^a^ (46.7)	20 ^b^ (83.3)	15 ^a^ (55.6)	70 ^b^ (85.4)	
Yes	44 (27.0)	16 ^a^ (53.3)	4 ^b^ (16.7)	12 ^a^ (44.4)	12 ^b^ (14.6)	
**hrHPV**						0.538
Negative	110 (67.5)	22 (73.3)	17 (70.8)	20 (74.1)	51 (62.2)	
Positive	53 (32.5)	8 (26.7)	7 (29.2)	7 (25.9)	31 (37.8)	

LVI: lymphovascular invasion, NI: nerve invasion; Significant values (*p* < 0.05) are indicated as bold. Bonferroni correction. Each superscript letter (a or b) denotes a subset of mutation categories whose column proportions do not differ significantly from each other at the 0.05 level.

**Table 4 cancers-12-01796-t004:** Multivariate logistic regression analysis of immune clusters associated with LNM.

Variable	Contrast	Multivariable Analysis	
		HR [95% CI]	*p*
Tumor grade	Per grade	3.610 [1.815–7.181]	<0.001
T stage	Per T stage	1.908 [1.082–3.364]	0.026
LVI	Present vs. absent	41.248 [3.657–465.225]	0.003
Histological Subtypes	Related vs. Non-related	0.105 [0.016–0.664]	0.017
Immune clusters	A and C vs. B and D	2.482 [1.057–5.833]	0.037

LVI: lymphovascular invasion, HR: hazard ratio.

**Table 5 cancers-12-01796-t005:** Antibody characteristics and used protocol for whole tissue IHC.

Marker	Principal Role	Catalog Number	Company	Dilution	Positive Control	Cellular Localization
**CD8**	Cytotoxic T cell	ZA-0508	ZSJQ-BIO	1:200	Tonsil	Membrane/cytoplasm
**FOXP3**	Regulatory T cell	YT6169	Immunoway	1:100	Tonsil	Nucleus
**Granzyme B**	Cytotoxic T cell granules	ZA-0599	ZSJQ-BIO	1:200	Tonsil	Cytoplasm
**CD68**	Macrophage	ZM-0060	ZSJQ-BIO	1:200	Tonsil	Cytoplasm
**CD206**	M2 macrophage	Ab64693	Abcam	1:100	Lung	Membrane/cytoplasm
**PD-1**	Immune checkpoint	ZM-0381	ZSJQ-BIO	1:200	Tonsil	Cytoplasm/membrane
**CTLA-4**	Immune checkpoint	Ab227709	Abcam	1:100	Tonsil	Membrane/cytoplasm
**PD-L1**	Immune checkpoint	13684(E1L3N)	Cell Signaling	1:400	Lung carcinoma	Membrane/cytoplasm
**Siglec-15**	Immune checkpoint	Ab198684	Abcam	1:100	Tonsil	Cytoplasm

## Data Availability

The authenticity of this article has been validated by uploading the key raw data onto the Research Data Deposit public platform (www.researchdata.org.cn), with the approval RDD number RDDB2020000845.

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
