# Peer review of "Immunophenotypes Based on the Tumor Immune Microenvironment Allow for Unsupervised Penile Cancer Patient Stratification"

_cancers, 2020, doi:10.3390/cancers12071796_

Round 1

Reviewer 1 Report

In general, the present study describes the Tumor associated microenvironmental immunophenotypes of peSCC. The results are useful for prediction of tumor immune escape from survaillence and clinical application via understanding of  immune-associated antigens. The authors have examined the antigen expressions such as CD8, GrB, FOXP3, CD68, CD206, PD-L1, CTLA-4, and Siglec-15 in the SCC patients. In addition, TIME-HPV axis has been discussed for clinical immunophenotype significance.

Some major comments are:

The immunohistochemistry results should be reproducible and the exmination number should be increased to justify the outcomes. This is because the factors examined are frequently expressed on the clinical tissue samples.

Siglec-15 expression should be corelated with the counterpart's sialic acid expression on cell surface. Normally tumor cells express Siglec-15 or other ITIM-bearing inhibitory receptors. Therefore, the Siglec-15 downstream molecules can be traced to justify the significance.

The study is weak in its impact ue to the above limitation.

In abstract, hrHPV should be abbreviated. 

Reviewer 2 Report

Most of the observations have been addressed by the authors. The micrographs and tables have importantly improved the readability of the paper. 

Author Response

Most of the observations have been addressed by the authors. The micrographs and tables have importantly improved the readability of the paper. 

We thank the reviewer for considering our study positively and for providing positive feedback and comments.

This manuscript is a resubmission of an earlier submission. The following is a list of the peer review reports and author responses from that submission.

Round 1

Reviewer 1 Report

  • The manuscript should go through extensive editing for grammar mistakes, particularly the introduction section.
  • Lines 21-23: This sentence is grammatically incorrect as it lacks a subject. I assume the authors meant “The aim of this study was to…”
  • Line 27: Phrase “associated with prognosis” needs clarification; LNM used as an abbreviation without previously having the term written in full.
  • Results section is difficult to follow as it lacks some general concluding remarks on the figures; specifically under the heading Immunophenotypes associated with prognosis. Perhaps the reader would benefit from a simplified graphical scheme of the clusters.
  • A co-staining for one of the relevant immune checkpoint markers and a specific macrophage/T cell marker could be provided for further validation.
  • The results section also sounds more like a method section, particularly under heading Association of immunophenotypes with lymph node metastasis, because of a lack of a short summary/explanation/concluding ending of a paragraph.

Reviewer 2 Report

The tumor-associated microenvironment and immune microenvironment (TIME) are important as penile squamous cell carcinoma (peSCC) may also exhibit the similar pathogenic roles. The authors examine any importance of the TIME with the high-risk human papillomavirus (hrHPV) infection. Then, they have examined the stromal and intratumoral CD8, granzyme B, FOXP3, CD68, CD206, PD-1, PD-L1, CTLA-4, and Siglec-15 synthesis by immunohistochemical staining of peSCC. Several GrB, FOXP3, CD68, CD206, PD-1 and CTLA-4 are related to DSS. The diffuse PD-L1 tumor cells are linked to prognosis than marginal/negative PD-L1 synthesis. Likely, peSCC immunophenotypes can be used as prediction markers of LNM and prognosis n stratification or immunotherapeutic designation.

Unfortunately, the present study is not conclusive for the phenotypes in their combinations. Several randomized results are not well presented. In addition, the results should be forwarded to such clinical case reports, not to the concise forms.

Criticisms are:

The patients (178 invasive peSCC patients (median age 52 years, range 24–86 years) are not classified in their male or female.

Fig. 1 is not presented. The regular figures are not presented due to some unexpected reasons during uploading step.

Table S1. Several biomarkers such as PD-1, CTLA-4, …… and etc have been tried to examine but the not direct relationship between the papilloma virus-positiveness and the peSCC, if applicable to other cancers, is perspective or discussed from the real results. Especially, Siglec-15 has been mentioned but no any relevant results. The Supplementary Table S1 presents the Siglec15 but the relationship between the hrHPV- + is not meaningful in the pathology. In addition, the HPV-infected cellular transformation does not induce the abnormal sugar synthesis. If the Sialic acid-alpha2,3-Gal-in N-glycans are documented, the pervious rationale should be described in the purpose section of the study.

Fig. S1. S2: The direct relationships between the parameters are not clearly applicable with their weak meanings.

Reviewer 3 Report

Chu et al. have used digital pathology analysis of multiple immune cell markers in peSCC to characterise the tumour immune microenvironment. This analysis identifies increased stromal GZMB and tumoural-PD1 expression in high risk HPV+ compared to hrHPV- samples. Multiple immune cell markers are found associated with improved survival rates and diffuse PD-L1 expression in the tumour is found associated with poor survival rates compared to marginal PD-L1 expression. Clustering analysis of immune cell markers identified 4 peSCC immunophenotypes, which were found to be associated with disease specific survival rates and disease stage.

This work provides a fairly comprehensive analysis of immune markers in a relatively poorly characterised cancer type. The use of quantitative digital pathology in this study is excellent, providing precise insight into the variation in immune cell markers between peSCC samples. Furthermore, the link identified between these features and disease stage/survival may provide useful insight for future studies examining the mechanisms of peSCC progression. However, there are some minor issues that should be revised prior to publication, detailed below.

  • As highlighted by the authors in the discussion, a limitation to this study is the lack of a validation dataset. If it is not possible to attain access to a second cohort to validate the survival correlations presented, efforts should be made to use a statistical approach less susceptible to over-fitting. The survival correlations presented in Fig. 4 were generated from the optimal cut-point. Thereby assessing the best-possible division of patients in this single dataset. Instead or in addition to this approach the authors should assess the immune markers association with survival as continuous variables (rather than categorical) using cox regression modelling.
  • The final finding that immunophenotype clusters and disease stage are linked is very interesting. This should be displayed in a main figure (g. as a spineplot as per figure 3).
  • Throughout the manuscript the text/grammar could be improved. Also, there seems to be some typos or wording choices that impact the interpretation. A non-exhaustive list is provided below:
    • Line 54. Presumably should read “hrHPV+ tumours have a survival advantage”
    • Line 82. Poor grammar at the start of this paragraph.
    • Line 203. Refers to “significant differences” but no statistical tests are presented comparing stromal and intratumoral compartments.
    • Line 328. Cluster B is named “immune resistance”. This suggests that the tumours in this cluster are resistant to the immune response. Perhaps an alternative name could be “inflammatory” to provide consistency with pan-cancer immune classification of tumours with high CD8 infiltration and improved survival rates.
  • Presentation of the data could also be improved to facilitate interpretation. Specific examples:
    • In Fig.4a presenting the HR plots on a log2 scale so that the positive and negative HRs scale equally.
    • In Fig.5b Scaling the values in the heatmap by row could make visualising differences between cluster easier.

Reviewer 4 Report

General comments and abstract

  • The manuscript could benefit from editing for grammar, missing words and subject-verb agreement, etc. It is recommended that authors delete irrelevant "general" phrases and sentences, repeated and unneeded words. They should use short sentences. Also, some Introductory sentences are irrelevant or are not needed. It is also recommended that authors send their manuscript to an expert in English editing and academic writing. For example in the abstract, authors stated “To examine the stromal 21 and intratumoral expression of CD8, granzyme B, FOXP3, CD68, CD206, PD-1, PD-L1, CTLA-4, and 22 Siglec-15 with immunohistochemistry in 178 samples of peSCC.” This is an incomplete sentence.
  • The abstract is poorly written. Why did authors choose to examine the stromal and intratumoral expression of CD8, granzyme B, FOXP3, CD68, CD206, PD-1, PD-L1, CTLA-4, and Siglec-15? What is their relation to the TIME? Those questions should be answered.
  • All abbreviations should be revised and defined at their first use, such as GrB and DSS in the abstract.
  • All the figure legends should be revised as to be more informative of the main results presented. 
  • Figures 1 and 2: add labels on the figures to specify the marker stained.
  • Figures 2c and 2d are the same with different magnifications. However, in the text, authors stated “There were 156 out of 170 cases with Siglec-15 positive expression in the stroma of the tissues, including 130 (76.5%) cases exhibiting low expression (Score 1-3) and 26 (15.3%) exhibiting high expression (Score 4) (Fig. 2c, d).” Kindly include an image showing low expression vs. high expression of Siglec-15 and add labels on the figures.

Introduction

  • Some references that the authors have used might not be up to date. Authors are kindly asked to review the whole manuscript and check the references accordingly. For example, the first statement used a reference [1] going back to 2015 while they could have used data from ‘Cancer Statistics by Siegel et al., 2020’ (https://doi.org/10.3322/caac.21590).
  • Authors stated “In-depth understanding of the TIME could better stratify the prognosis of cancer patients …” this statement should be written as “better stratify cancer patients according to their prognosis”
  • “lower rate of LNM” This statement should be fixed as LNM staging has no rates but stages.
  • Authors elaborated extensively on the role of some TIME markers, such as PD-L1, CTLA-4, CD68, GrB, and CD206. I believe it would be more appropriate to move some details from the introduction and include them in the discussion.
  • “TIME was classified into three basic types: immune inflamed, immune desert and immune excluded.” In what tumor type? And kindly add the citation for the statement.
  • What do authors mean by “unsupervised clustering analysis”? This point needs to be clarified.

Methods

  • Kindly add the catalogue numbers of the different antibodies used (Table 1). Also, I suggest that authors specify whether each marker is a nuclear, cytoplasmic, or membrane stain.
  • Authors explained how PD-L1 and Siglec-15 expressions were assessed. However, they did not specify how other IHC markers were assessed.
  • The statistics section is very well explained.

Results and Discussion

  • Results and Discussion sections are very well written.

Reviewer 5 Report

Chu and colleagues, propose a stratification for patients with penile cancer in 4 Clusters. The grouping is based on the characterization of a list of immunomarker taken from the literature. This is the first report for a set of immunomarker used to individuate the TIME for penile cancer. The problem is well introduced and addressed with unsupervised techniques that avoid misinterpretation of the data. However, there are some concerns.

Major points:

  • Micrographs of all the immunomarkers used in the study have to be shown as for PD-1 and CD8, to demonstrate the correct localization and staining of the protein.
  • Siglec-15 is a marker of worse prognosis as cited by the authors. However, in Fig.4, high levels of the protein in the stroma are associated with better outcomes. How do the authors explain this discrepancy?
  • Another marker for M2 macrophages should be used to better explain the results of cluster C. The authors mention the CD163 marker that could be used.
  • A table with the role (in tumor progression) of the immunomarker used, and the class of inflammatory cells represented, would help in comparing data in the literature with the stratification in clusters.
  • Also, a table describing the immunomarker expressed in different clusters A-D would give a more direct message to the reader.
  • 3a is not described in the text. Add and comment on the figure.
  • Micrographs with staining representing the unsupervised stratification should be shown to assess the difference between low and high expressing tumors/stroma.
  • Provide more details in the figure legends.

Minor points:

  • Not all the figures are cited in the text (i.e. fig. 3a), or they are not cited in the correct order (i.e. fig.6 cited before 5 c-e). Adjust all over the text.
  • Add details in all figures with IHC micrographs (i.e. in fig. 1 indicate which panel is CD-8 and PD-1). Also figures number is missing throughout the paper.
  • 2a and b are inverted in the text. Adjust accordingly with figure legend. Moreover, fig. 2d is a magnification of fig. 2 c. Add this information to the text. Also in fig legend magnification (x200) should be adjusted to the correct magnification.